# The impact of the strategic action plan to combat chronic non-communicable diseases on hospital admissions and deaths from cardiovascular diseases in Brazil

**Rafaella Alves da Silva** [1]*, **Luiza Gabriela de Araújo Fonseca**[1], **João Pedro de Santana Silva**[2], **Núbia Maria Freire Vieira Lima**[1], **Lucien Peroni Gualdi**[1], **Illia Nadinne Dantas Florentino Lima**[1]

**1** Faculdade de Ciências da Saúde do Trairi (FACISA)/Universidade Federal do Rio Grande do Norte (UFRN), Programa de Pós-graduação em Ciências da Reabilitação, Santa Cruz, Rio Grande do Norte, Brazil, **2** Curso de Fisioterapia, Faculdade de Ciências da Saúde do Trairi (FACISA)/Universidade Federal do Rio Grande do Norte (UFRN), Santa Cruz, Rio Grande do Norte, Brazil

* rafaellaalves.fisio@gmail.com

**Data Availability Statement:** All relevant data are within the paper.

## Abstract

Chronic Non-Communicable Diseases (NCDs) are the main causes of death worldwide, responsible for millions of hospital admissions per year, especially cardiovascular diseases (CVD). Several strategies for controlling and coping with these diseases have been developed in several countries. The aim of the study was to evaluate the impact of the Strategic Action Plan to Combat NCDs (2011–2022) on hospital admissions, deaths and mortality rate in Brazil, classified by CVD. This is a descriptive study, with secondary data from the Hospital Information System of the Unified Health System (SIH/SUS). Hospital admissions, deaths and mortality rate due to CVD in the Brazilian population aged over 20 years were analyzed, according to region, sex and age group. Statistical analysis was performed using the GraphPad Prism program. Data normality was assessed using the Komogorov Smirnov test and the comparison between groups and year periods was performed using the two-way ANOVA test with Tukey's post hoc test. A value of p<0.05 was considered significant. In this study, in most analyses, a reduction in the hospitalization rates of the adult population was observed after the implementation of the plan, however, there was no improvement in relation to the number of deaths and mortality rate from CVD. This shows that there is still a long way to go to reduce the impact of these diseases in Brazil, and they reaffirm the need for and importance of maintaining the prevention of their risk factors, the social determinants of health and the reorganization of care in the face of to population aging. Such findings contribute with information that allow better control and monitoring of CVD and should be considered when implementing new strategies for prevention, care and control of risk factors.

**Funding:** The authors received no specific funding for this work.

**Competing interests:** The authors have declared that no competing interests exist.

## Introduction

Chronic Non-Communicable Diseases (NCDs) represent one of the major public health challenges in recent decades. They are characterized by having an uncertain etiology, non-infectious origin, tend to be of long duration, and result from a combination of genetic, physiological, environmental and behavioral factors [1]. They are often associated with impairments and functional incapacities, in addition to having long latency periods and multiple risk factors [2]. According to the World Health Organization (WHO), NCDs are responsible for 41 million deaths per year, equivalent to 71% of all deaths in the world, especially cardiovascular diseases (CVDs), followed by cancer, respiratory diseases and diabetes mellitus, which are responsible for more than 80% of premature deaths (30 to 69 years old) from NCDs (1). They correspond to 72% of all causes of death in Brazil, with 54.7% of deaths from NCDs being identified in 2018, and 11.5% from their complications [3,4].

CVDs are responsible for the majority of deaths from NCDs, generating approximately 17.9 million deaths annually [1]. According to the WHO, an increase in CVD cases has been observed, mainly in low- and middle-income countries, reflecting the difficulty in accessing effective health services for their needs, often resulting in late diagnosis and early deaths. In addition, they contribute to world poverty due to high health expenditures, creating a heavy burden on the economies of these countries [5].

CVDs were responsible for 18.71% of deaths in Brazil until 2019, being considered the main cause of death, in addition to being the factor with the greatest impact on the cost of hospital admissions in the country. Hospital admissions resulting from CVD for that same year represented 9.68% and of the total of these hospital admissions, and 25.46% were among individuals aged 60 to 69 years [6]. According to data from the Brazilian Institute of Geography and Statistics (*IBGE*), the aging of society is evident, increasing the proportion of older adults and life expectancy [7]. Physiological and behavioral changes related to aging tend to increase the incidence of CVD in older adults, and consequently their health costs [8].

Public policies in recent times have turned attention to expanding programs which encourage adopting an active and healthy lifestyle [9]. Clarification regarding health conditions and the importance of medical care are fundamental for organizing strategies for promotion, prevention and healthcare [10]. Establishing concrete actions to fight these diseases is essential to reduce mortality and costs, in addition to contributing to improve the quality of life of patients [2].

In this context, a High Level Declaration was established in 2011 at the United Nations with the objective of reducing NCD mortality rates [11]. The Ministry of Health in Brazil launched the Action Plan to Combat NCDs (2011–2022) in 2011, with the objective of promoting the development and implementation of public policies to control these diseases and their risk factors. The plan addresses the four main groups of NCDs (circulatory, cancers, chronic respiratory and diabetes), as well as their common and modifiable risk factors: smoking, alcohol, physical inactivity, unhealthy diet and obesity [12].

The Plan is based on three main guidelines and axes: I) Surveillance, information, evaluation and monitoring; II) Health promotion; and III) Comprehensive care. Its main objective is to reduce early mortality (30 to 69 years) by 25% by 2025, as well as to reduce risk factors [13]. Studies after three years of its implementation show that mortality reduction targets have already been achieved for the entire country, and a significant reduction of its risk factors [14].

The implementation strategies presented in the plan involved general actions for adherence to healthy eating, encouraging breastfeeding, food labeling and agreements with industries to eliminate fat and reduce salt in foods. The implementation of physical activity took place through the Healthy academy program, with a goal of implementation in more than 4,000

municipalities by 2015. To prevent tobacco use, regulatory actions were established for advertising and selling cigarettes and alcohol. And finally, free distribution strategies for more than 15 types of drugs for hypertension and diabetes [12,13].

Therefore, this study aims to analyze the impact of the Strategic Action Plan to Combat NCDs from 2011–2022 on hospital admissions, deaths and mortality rate in Brazil classified by CVDs, to analyze the incidence and regional distribution, as well as the time trend from 2008 to 2019.

## Methodology

### Study characterization

This is a descriptive, longitudinal, quantitative study. Data collection was performed from hospital admissions registered in the Unified Health System (*SIH/SUS*) classified by CVD according to ICD 10 (International Classification of Diseases) from 2008 to 2019. Hospital admissions, deaths and mortality rate due to CVDs in the Brazilian population, aged over 20 years, with a national and regional focus were analyzed. All records in public or private services transmitted to *DATASUS* were accessed and included in the study.

### Ethical aspects

Ethical approval is not required for the present study according to the National Health Council that regulates the National Research Ethics Committee (*CONEP*), Resolution No. 510 of April 7, 2016. All data are public and of free access via *DATASUS* (http://datasus.saude.gov.br/).

### Data extraction

Data were taken from the *SIH/SUS*, provided by the Health Surveillance Secretariat of the Ministry of Health through the open access electronic portal available at *DATASUS* (http://datasus.saude.gov.br/). The following variables were extracted: number of hospital admissions, deaths and mortality rate from CVDs as classified by the ICD-10. Absolute values and frequencies were grouped according to gender, age group (20–29; 30–39; 40–49; 50–59; 60–69; 70–79; 80 years and over), region of residence, morbidity list and year of processing (2008–2019). The number of hospital admissions and deaths was calculated by the ratio between the number of hospital admissions and deaths recorded and the estimate of each region Brazilian population, according to the IBGE. All data were extracted in April 2021, grouped and stored in spreadsheets in the Microsoft Excel version 2019 program for further statistical analysis.

### Statistical analysis

Statistical analysis was performed using the GraphPad Prism version 7.0 program. The sample normality was analyzed using the Kolmogorov Smirnov test. A two-way ANOVA test with Tukey's post hoc test was performed for comparison between groups and periods of the year. A value of $p < 0.05$ was considered significant. The complete analysis can be found in the S1 File.

## Results

### Hospital admissions due to CVDs

A total of 13,380,119 hospital admissions for CVDs in Brazil were recorded in the *SIH/SUS* in adults aged 20 years and older during the period from 2008 to 2019. The highest hospital admission incidence was due to a diagnosis of Heart Failure (20.68%), other ischemic heart

**Table 1. Relative and absolute frequency of hospital admissions for cardiovascular diseases according to morbidity and quadrennium (2008 to 2011; 2012 to 2015; 2016 to 2019).**

| Morbidity | 2008 to 2011 | 2012 to 2015 | 2016 to 2019 |
|---|---|---|---|
| **Unspecified Cerebral Vascular Accident** | **438020 (9.9%)** | **545656 (12.3%)** | **619059 (13.8%)** |
| Atherosclerosis | 52935 (1.2%) | 64125 (1.4%) | 91038 (2%) |
| Chronic rheumatic heart disease | 27538 (0.6%) | 31327 (0.7%) | 27841 (0.6%) |
| Arterial embolism and thrombosis | 63657 (1.4%) | 75715 (1.7%) | 85485 (1.9%) |
| Pulmonary embolism | 20038 (0.5%) | 25617 (0.6%) | 34682 (0.8%) |
| Acute rheumatic fever | 16196 (0.4%) | 12643 (0.3%) | 6701 (0.1%) |
| Phlebitis thrombophlebitis embolism and venous thrombosis | 134547 (3%) | 159083 (3.6%) | 165068 (3.7%) |
| Intracranial hemorrhage | 143310 (3.2%) | 110196 (2.5%) | 107061 (2.4%) |
| Hemorrhoids | 122122 (2.7%) | 110293 (2.5%) | 109915 (2.4%) |
| Essential (primary) hypertension | 387619 (8.7%) | 303385 (6.8%) | 221582 (4.9%) |
| Acute myocardial infarction | 285333 (6.4%) | 366099 (8.2%) | 469097 (10.4%) |
| Cerebral infarction | 56374 (1.3%) | 62005 (1.4%) | 83180 (1.9%) |
| **Cardiac insufficiency** | **1048338 (23.6%)** | **908414 (20.5%)** | **810241 (18%)** |
| Other cerebrovascular diseases | 58896 (1.3%) | 64501 (1.5%) | 66757 (1.5%) |
| Other diseases of arterioles and capillaries | 146534 (3.3%) | 122046 (2.7%) | 127249 (2.8%) |
| Other diseases of the circulatory system | 35012 (0.8%) | 29536 (0.7%) | 32680 (0.7%) |
| Other heart diseases | 134135 (3%) | 126994 (2.9%) | 129234 (2.9%) |
| Other hypertensive diseases | 140265 (3.2%) | 105178 (2.4%) | 84575 (1.9%) |
| **Other ischemic heart diseases** | **570906 (12.8%)** | **625272 (14.1%)** | **615693 (13.7%)** |
| Other peripheral vascular diseases | 25812 (0.6%) | 27465 (0.6%) | 43651 (1%) |
| Conduction disorders and cardiac arrhythmias | 199794 (4.5%) | 229773 (5.2%) | 249727 (5.6%) |
| Varicose veins of the lower extremities | 339361 (7.6%) | 335683 (7.6%) | 311855 (6.9%) |
| **Total** | **4446742** | **4441006** | **4492371** |

Source: Ministry of Health–Hospital Information System of the Unified Health System (SIH/SUS) 2021.

diseases (13.54%) and stroke (11.98%). The hospital admissions for cardiovascular diseases according to morbidity and quadrennium was presented in Table 1.

**Hospital admissions per 100,000 inhabitants according to region and year.** When evaluating the difference between region of residence and year in relation to hospital admissions/100 thousand inhabitants, it is possible to observe that there is a difference for region (p<0.001) and year (p<0.001). When grouping the averages of hospital admissions according to region, it was found that the highest incidence was in the South region (831.38), followed by the Southeast region (587.52), Midwest (517.58), Northeast (439.75) and North (319.53), with a difference between all groups studied (p<0.001), according to Fig 1.

It is noted that there have been variable changes over the years, but a decrease in the amplitude of growth is observed despite the annual increase. According to the averages, the highest prevalence of hospital admissions was in 2009 (569.71), followed by 2010 (565.78), 2011 (561.26) and 2008 (556.37). This same predominance is observed when evaluating the difference between region of residence and four-year period per 100,000 inhabitants (Fig 2), showing a higher incidence in 2008–2011 (563.23), followed by 2012–2015 (533.37) and 2016–2019 (518.86).

There was a reduction in hospital admissions of 5.3% between the 2008–2011 and 2012–2015 quadrennia (p<0.001), and of 7.88% between 2008–2011 and 2016–2019 (p<0.001). Despite the 2.72% reduction, there was no significant difference when comparing the 2012–2015 and 2016–2019 quadrennia (p>0.05).

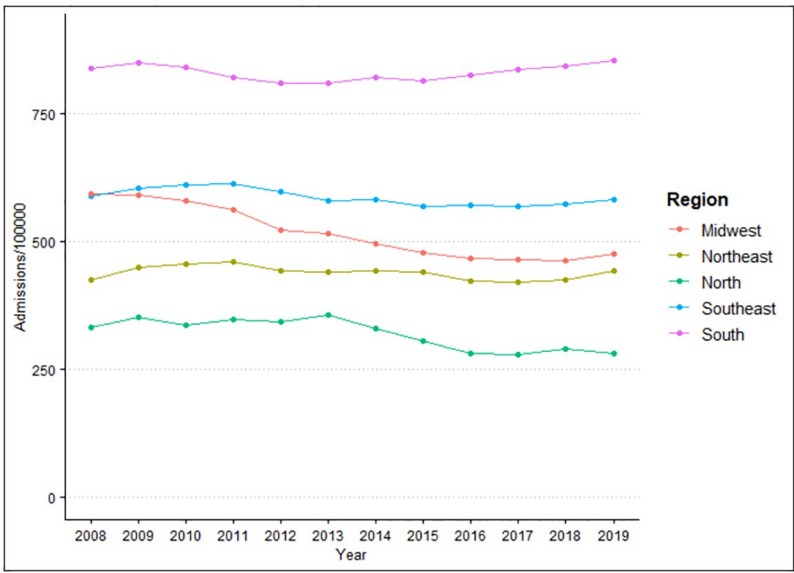

**Fig 1. Number of hospital admissions caused by cardiovascular diseases per 100,000 inhabitants according to region of residence in the period from 2008 to 2019.** Source: Brazilian Institute of Geography and Statistics (IBGE) and Ministry of Health–Hospital Information System of the Unified Health System (SIH/SUS) 2021.

**Hospital admissions according to age group and quadrennium.** When evaluating the difference between age group and four-year period, it is possible to observe that there is a difference for age group (p<0.001), but not for four-year period (p = 0.67). The hospital admission averages show a higher incidence in individuals aged 60 to 69 years (268,997.75), followed by 70 to 79 years (234,585.17), 50 to 59 years (224,233.08), 80 years and over (150,348, 25), 40 to 49 years (133,405.33), 30 to 39 years (69,388.33) and 20 to 29 years (34,052), with no statistical difference when comparing the groups 50 to 59 and 70 to 79 years (p. = 0.05). In analyzing

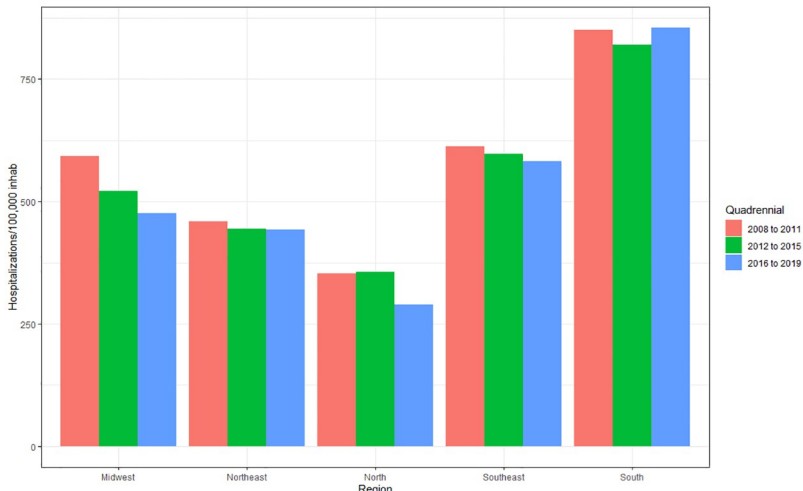

**Fig 2. Average of hospital admissions caused by cardiovascular diseases per 100 thousand inhabitants according to region of residence and quadrennium (2008 to 2011; 2012 to 2015; 2016 to 2019).** Source: Brazilian Institute of Geography and Statistics (IBGE) and Ministry of Health–Hospital Information System of the Unified Health System (SIH/SUS) 2021.

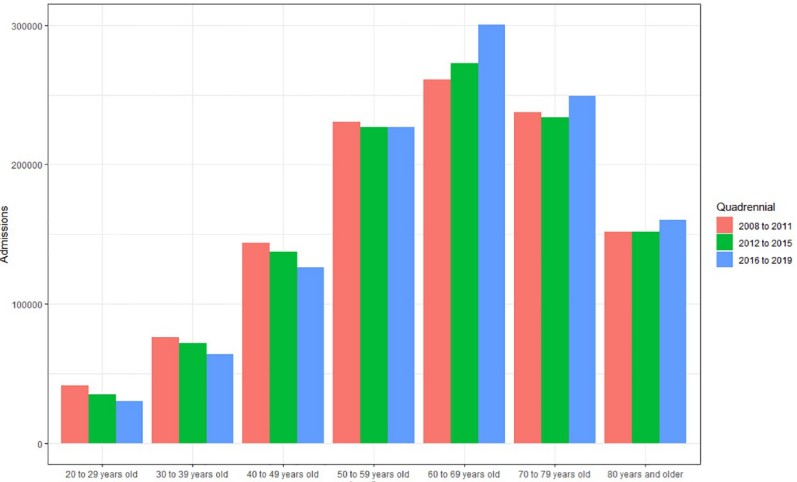

**Fig 3. Average of hospital admissions caused by cardiovascular diseases according to age group (>20 to 80 years and over) and quadrennium (2008 to 2011; 2012 to 2015; 2016 to 2019).** Source: Ministry of Health–Hospital Information System of the Unified Health System (SIH/SUS) 2021.

the four-year period, there is a decline in the hospitalization rate in individuals aged 20 to 59 years, whereas there is an increase for those aged over 60 years (Fig 3). The 60–69 age group showed an increase of 13.76% between 2008–2011 and 2016–2019.

**Hospital admissions according to gender and quadrennium.** In analyzing the difference between gender and quadrennium, it is noted that there is no difference for either gender (p>0.64) or quadrennium (p>0.62). Despite the small amplitude of growth over the years, it is possible to notice that female gender presented a reduction between the quadrennium's, while male gender presented a reduction followed by an increase (Fig 4). Females had a reduction of 2.72% between 2008–2011 and 2016–2019, and males had an increase of 4.89% between 2008–2011 and 2016–2019.

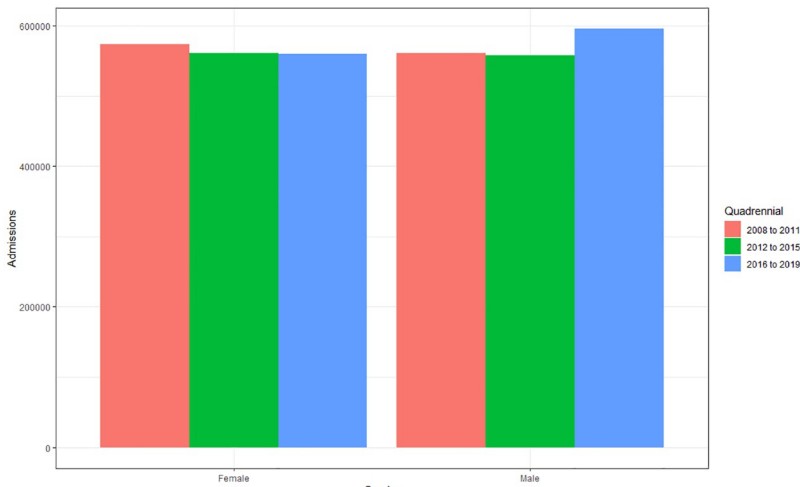

**Fig 4. Average of hospital admissions caused by cardiovascular diseases according to gender and quadrennium (2008 to 2011; 2012 to 2015; 2016 to 2019).** Source: Ministry of Health–Hospital Information System of the Unified Health System (SIH/SUS) 2021.

## Deaths by CVDs

There were 1,058,953 deaths recorded in the *SIH/SUS* due to CVDs in adults aged 20 years and over in Brazil during the period from 2008 to 2019. Regarding the diagnosis, the highest incidence of deaths was due to heart failure (25.6%), stroke (24.4%) and acute myocardial infarction (12.4%).

**Deaths per 100,000 inhabitants by region and year.**   After evaluating the difference between region of residence and year in relation to deaths/100 thousand inhabitants, it was observed that there is a difference both for region (p<0.001) and for year (p<0.001). When grouping the averages of deaths by region, it was found that the highest incidence was in the South region (54.04), followed by the Southeast region (49.42), Midwest (39.48), Northeast (36.56) and North (25.4), with a difference between all groups studied (p<0.001), shown in Fig 5.

It is noted that there is a great variation in the amplitude of growth, with a greater tendency towards an increase in deaths over the years. The highest incidences of deaths according to the averages were in 2019 (44.05), followed by 2016 (43.2), 2018 (42.31), 2015 (41.71), 2017 (41.42), 2011 (40.96), 2010 (40.77), 2013 (40.56), 2014 (39.99), 2012 (39.94), 2009 (39.53) and 2008 (37.31). The differences are better observed when evaluated by quadrennium (Fig 6), which shows a higher incidence in 2016–2019 (42.75), followed by 2012–2015 (40.55) and 2008–2011 (39.64).

The four-year period 2016–2019 is significantly higher compared to 2012–2015 (p<0.001) and 2008–2011 (p<0.001), but there is no difference when comparing 2012–2015 and 2008–2011 (p>0.24). There was a 7.84% increase in deaths between 2008–2011 and 2016–2019.

**Deaths by age group and quadrennium.**   In analyzing the difference between age group and four-year period, it appears that there is a difference for both age group (p<0.001) and four-year period (p<0.001). The averages of deaths point to a higher incidence in individuals aged 70 to 79 years (23,943.92), followed by 80 years and over (23,638.58), 60 to 69 years

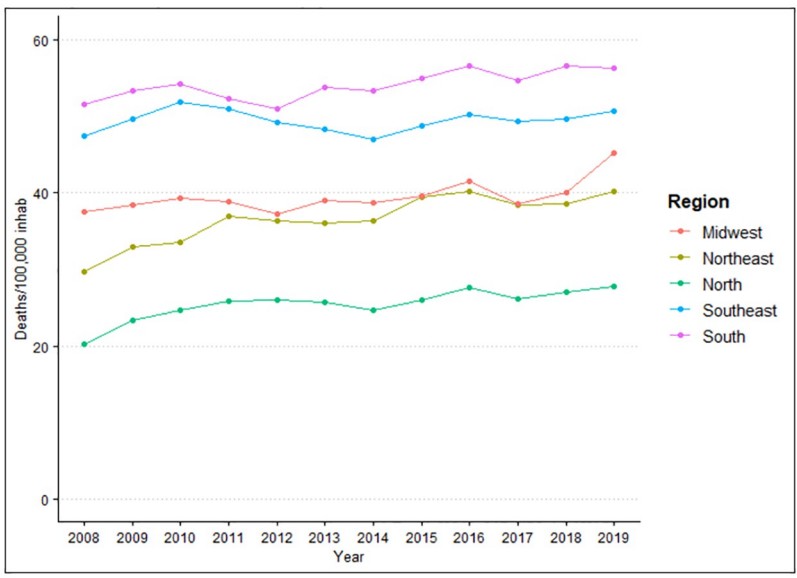

**Fig 5. Number of deaths caused by cardiovascular diseases per 100 thousand inhabitants according to the region of residence in the period from 2008 to 2019.** Source: Brazilian Institute of Geography and Statistics (IBGE) and Ministry of Health–Hospital Information System of the Unified Health System (SIH/SUS) 2021.

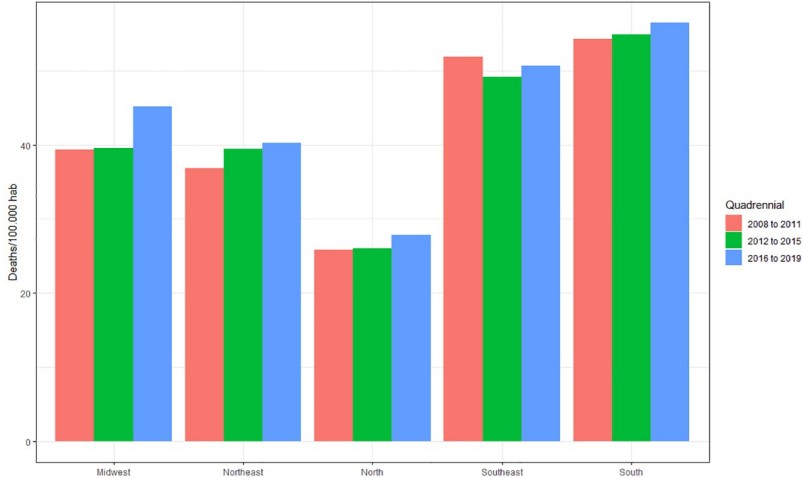

**Fig 6. Average of deaths caused by cardiovascular diseases per 100 thousand inhabitants according to region of residence and quadrennium (2008 to 2011; 2012 to 2015; 2016 to 2019).** Source: Brazilian Institute of Geography and Statistics (IBGE) and Ministry of Health–Hospital Information System of the Unified Health System (SIH/SUS) 2021.

(19,536.92), 50 to 59 years (12,127.08), 40 to 49 years (5,704), 30 to 39 years (2,271.83) and 20 to 29 years (1,023.75), with no statistical difference only in relation to the groups 20 to 29 years and 30 to 39 years (p>0.09), and the groups from 70 to 79 years and 80 years and over (p>0.99). There is a significant difference between age groups, with a greater predominance from 50 years of age, as well as the increase in deaths with advancing age, especially at the age of 70 years, which presented greater amplitude of growth with an increase of 28.23% 2008–2011 and 2016–2019 (Fig 7).

Regarding the four-year period, the predominance of deaths by age group was in 2016–2019 (13,476.86), after 2012–2015 (12,467.79) and 2008–2009 (11,875.11). The 2016–2019

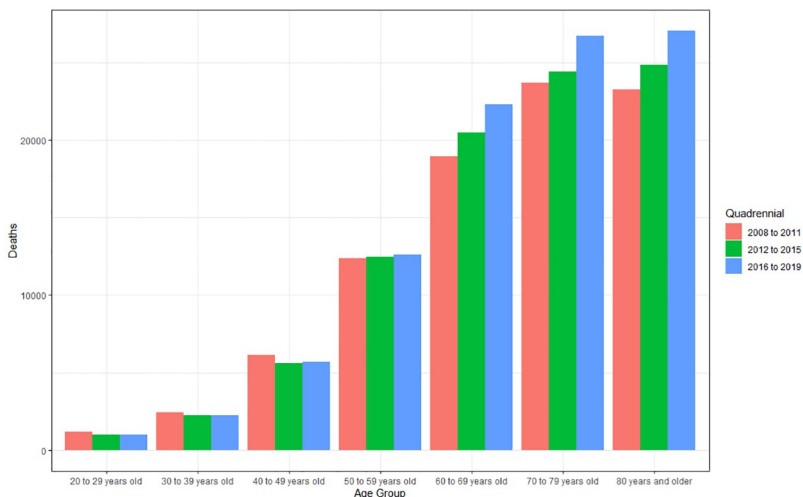

**Fig 7. Average of deaths caused by cardiovascular diseases according to age group (>20 to 80 years and over) and quadrennium (2008 to 2011; 2012 to 2015; 2016 to 2019).** Source: Ministry of Health–Hospital Information System of the Unified Health System (SIH/SUS) 2021.

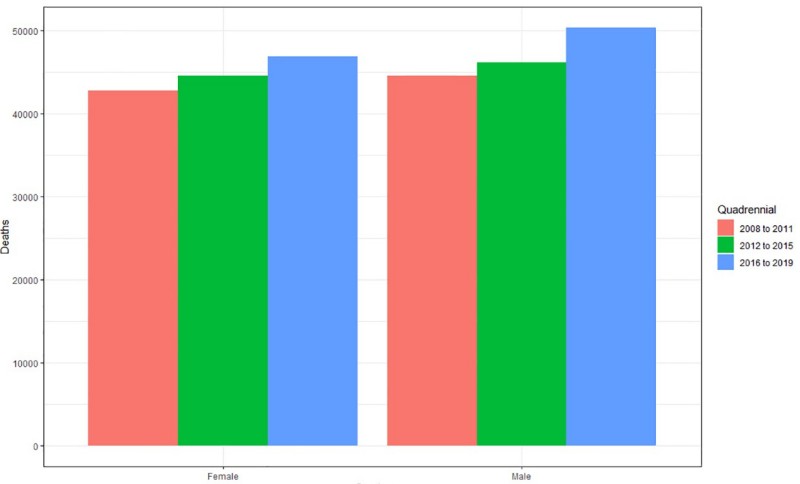

**Fig 8. Average of deaths caused by cardiovascular diseases according to gender and quadrennium (2008 to 2011; 2012 to 2015; 2016 to 2019).** Source: Ministry of Health–Hospital Information System of the Unified Health System (SIH/SUS) 2021.

quadrennium is significantly higher than 2008–2011 (p<0.001) and 2012–2015 (p = 0.003), but there is no difference when comparing 2012–2015 and 2008–2011 (p>0.11). There was an increase in the average number of deaths of 13.49% between 2008–2011 and 2016–2019.

**Deaths by gender and quadrennium.** Unlike hospital admissions, when analyzing the difference in deaths according to gender and quadrennium, it is noted that there is a difference both for gender (p = 0.03) and for quadrennium (p<0.001). According to the averages, the incidence of deaths was higher in males (45,188.92), which corresponds to 51.20% of deaths, followed by females (43,057.17), representing 48.79%, presenting an increase over the years (Fig 8). There was a 13.63% increase for males between 2008–2011 and 2016–2019, and a 13.34% increase for females between 2008–2011 and 2016–2019.

## Mortality by CVDs

**Mortality rate by region and year.** Note that there is a difference both for region (p<0.001) and for year (p<0.001) when comparing the mortality rate by region of residence and year. The data show the highest mortality rate in the Southeast region (8.1), followed by the Northeast region (8.32), North (8.13), Midwest (7.67) and South (6.5), according to Fig 9.

An increase in the most predominant growth amplitude can be seen in the North, Northeast and Midwest regions. On the other hand, despite an annual increase tendency in the Southeast and South regions, they showed a lower amplitude of growth. The mortality rate was higher in 2019 (8.78) than in 2016 (8.77), 2018 (8.48), 2017 (8.41), 2015 (8.2), 2014 (7.6), 2013 (7.58), 2012 (7.48), 2011 (7.4), 2010 (7.28), 2009 (6.99) and 2008 (6.72), confirming the sequential increase rate over the years.

These data are better identified when evaluated by four-year period (Fig 10), noting that the mortality rate was higher in 2016–2019 (8.61), followed by 2012–2015 (7.72), and finally in 2008–2011 (7, 1), with a significant difference in all groups studied (p<0.001 and p = 0.002).

**Mortality rate according to age group and quadrennium.** It is also possible to observe that there is a difference for age group (p<0.001) and for four-year period (p = 0.001) regarding the mortality rate. There was an increase in the mortality rate with increasing age, with the highest mortality rate in the age group of 80 years and over (15.59), followed by 70 to 79 years

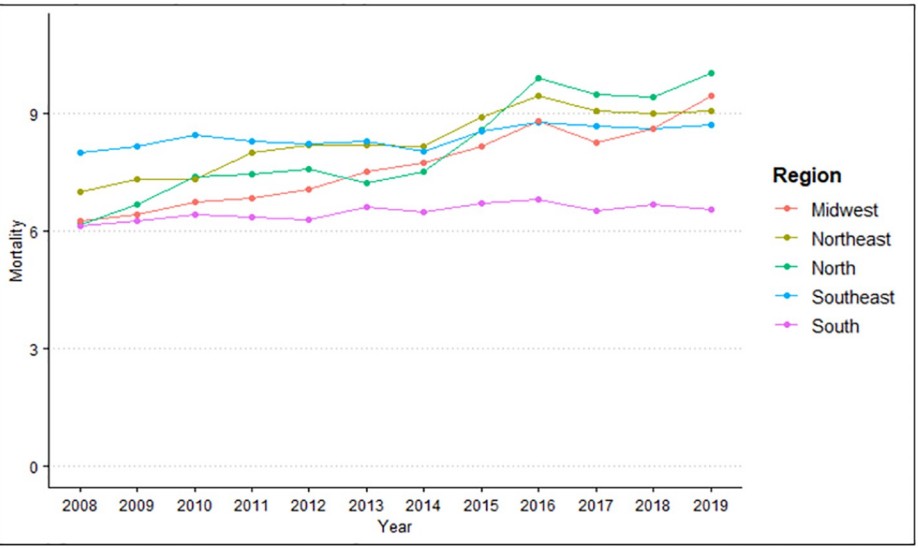

**Fig 9. Mortality rate from cardiovascular diseases according to region of residence in the period from 2008 to 2019.** Source: Ministry of Health–Hospital Information System of the Unified Health System (SIH/SUS) 2021.

(10.2), 60 to 69 years (7.25), 50 to 59 years (5.41), 40 to 49 years (4.28), 30 to 39 years (3.29) and 20 to 29 years (3.03), with no statistical difference only when comparing the groups 20 to 29 years and 30 to 39 years (p>0.64), as observed in Fig 11.

The highest rate according to age group was also in the four-year period 2016–2019 (7.4), followed by 2012–2015 (6.96) and 2008–2011 (6.7), with a significant difference between all of them (p<0.001 and p = 0.037).

**Mortality rate according to gender and quadrennium.** There is a difference both for gender (p = 0.002) and for quadrennium (p<0.001) regarding the mortality rate. The mortality rate was in agreement with the incidence of deaths, and was also predominant in males (8.08),

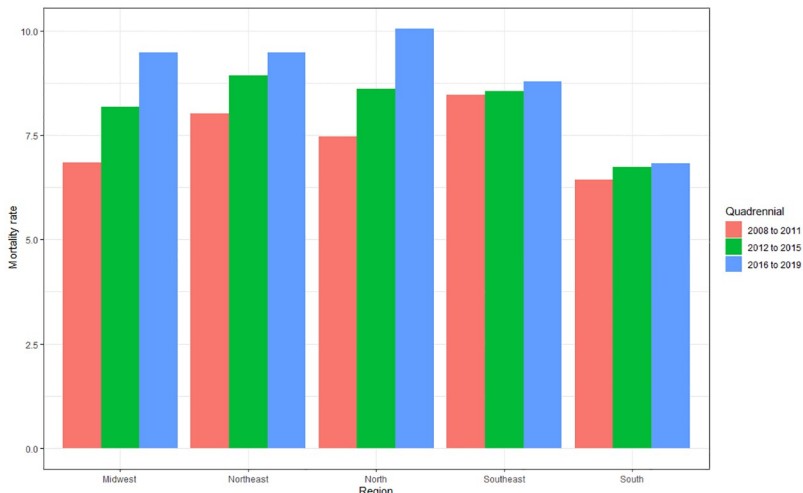

**Fig 10. Average mortality rate from cardiovascular diseases according to region of residence and quadrennium (2008 to 2011; 2012 to 2015; 2016 to 2019).** Source: Ministry of Health–Hospital Information System of the Unified Health System (SIH/SUS) 2021.

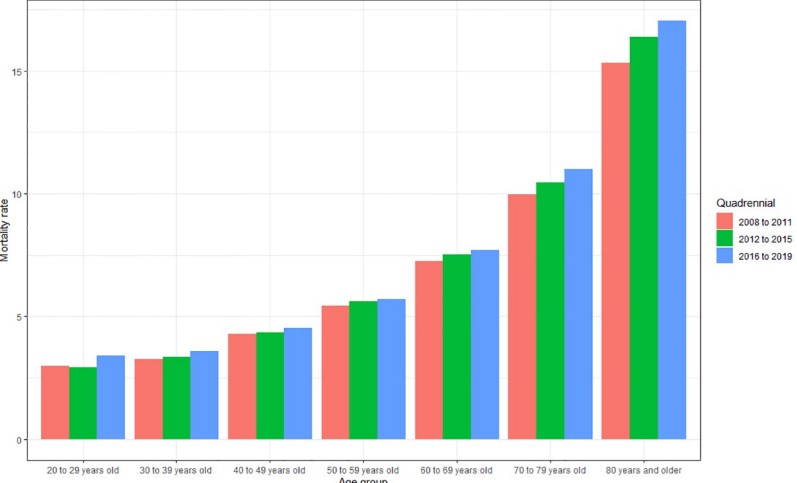

**Fig 11. Average mortality rate from cardiovascular diseases according to age group (>20 to 80 years and over) and quadrennium (2008 to 2011; 2012 to 2015; 2016 to 2019).** Source: Ministry of Health–Hospital Information System of the Unified Health System (SIH/SUS) 2021.

followed by females (7.75). For the four-year period, the predominance in the sequence 2016–2019 (8.4), 2012–2015 (7.89) and 2008–2011 (7.48) was also observed, Fig 12.

## Discussion

The present study highlights the record of 13,380,119 hospital admissions for CVDs in adults in Brazil from 2008 to 2019, with HF being responsible for the highest incidence with 20.68% of total hospitalizations. However, when the temporal analysis was performed, there was a 25.57% reduction in the number of hospitalizations for HF, from 264,103 in 2008 to 196,558 in 2019. This finding corroborates the study by Fernandes et al. (2020), who observed that HF

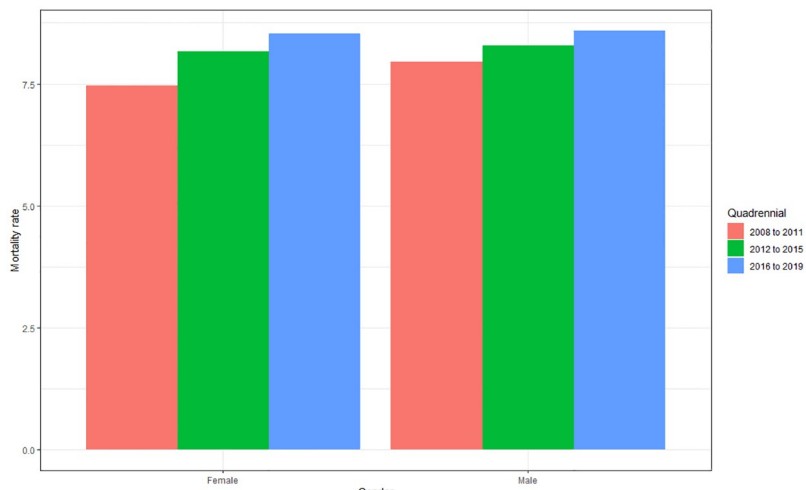

**Fig 12. Average mortality rate from cardiovascular diseases according to gender and quadrennium (2008 to 2011; 2012 to 2015; 2016 to 2019).** Source: Ministry of Health–Hospital Information System of the Unified Health System (SIH/SUS) 2021.

was also the main cause of hospitalizations for CVD in Brazil from 2008 to 2017, accounting for 21% of cases [15].

The main findings in the study with regard to hospital admissions for CVDs after implementing the Strategic Action Plan for Coping with NCDs (2011–2022) were: 1) a 25.77% reduction in hospital admissions for HF; 2) an increase of 0.25% in the hospitalization rate in the South region per 100 thousand inhabitants, while the other regions showed a reduction; 3) a 7.88% reduction in hospital admissions between 2008–2011 and 2016–2019; 4) a reduction in the hospitalization rate among individuals aged 20 to 59 years, and an increase among those aged 60 to 80 years and over, with greater amplitude of growth in the group aged 60 to 69 years (13.77%); and 5) a reduction of 2.72% of hospitalizations in females and an increase of 4.89% in males.

The averages of hospital admissions show a higher hospitalization rate in the South region with 831/100 thousand inhabitants, with an increase of 0.25% between 2008–2011 and 2016–2019. This expressive rate found in the South region may be related to the greater number of hospital beds, since this region has the highest rate of beds per inhabitant in Brazil [16]. In this study, the Midwest region showed the greatest downward trend in hospitalization rates when compared to other Brazilian regions. In another study, Santos et al. (2015) showed stability in the years 2002 to 2012 for CVD hospitalization rates in Brazil for all regions, except for the Midwest region, which was the only one that showed an annual decline of 8.78% [17].

For the temporal analysis, the highest prevalence of hospitalizations for CVDs was found in 2009 with 569.7/100 thousand inhabitants, and the lowest with 514.19/100 thousand inhabitants in 2017. The analysis by quadrennium shows a higher incidence in 2008–2011 with an average of 563.23/100 thousand inhabitants, with a decline of 7.88% for 2016–2019 with 518.86/100 thousand inhabitants. Such findings are similar to the study by Siqueira et al. (2017), which shows a drop in the number of hospitalizations for CVDs from 874,949 in 2010 to 807,304 in 2015, corresponding to a reduction of 7.73% [7].

According to the study by Santos et al. (2015), the highest hospitalization rates occurred in groups over 40 years of age, and especially in individuals over 70 years of age [18]. These data are consistent with the present study, which showed a higher hospitalization rate in individuals over 50 years of age, especially in the group aged 60 to 69 years, who were responsible for 24.13% of hospitalizations for CVD. In addition, the same group showed an increase of 13.77% between the 2008–2011 and 2016–2019 quadrennia.

The highest prevalence of hospitalizations regarding gender was in males (50.12%) when compared to females (49.88%). Although no significant differences were found between genders and the four-year period, females showed a reduction of 2.72% between 2008–2011 and 2016–2019, while males had an increase of 4.89%. Studies carried out in Brazil have found that men are more likely to have two or more risk factors for CVD, and that this increase is directly associated with age and lower education [19,20]. In addition, others also defend the fact that women use health services in a preventive way more frequently, favoring greater possibilities of early diagnosis [21,22].

Regarding deaths, 1,058,953 deaths from CVD were recorded in adults in Brazil during the period from 2008 to 2019, with a national mortality rate of 7.91. HF was also responsible for the highest incidence of 25.70% of total deaths; however, although the data show similarities over the years, there was a reduction of 6.46% between the 2008–2011 and 2016–2019 quadrennia.

There was an increase in the average of deaths in all regions between 2008 and 2019, with the highest average found in the South region with 54.04/100 thousand inhabitants and Southeast 49.42/100 thousand inhabitants, and the lowest in the North region with 25.4/100 thousand inhabitants. These data corroborate the data from the study by Oliveira et al. (2020),

which show that CVDs accounted for 27.3% of total deaths in 2017, with the highest proportion in the Southeast region and the lowest in the North region [23]. The mortality rate in this study also showed a progressive increase in all regions as the years went by, with a greater total predominance in the Southeast region (8.41).

The temporal analysis shows a higher prevalence of deaths from CVDs in 2019 with 44.01/100 thousand inhabitants and a mortality rate of 8.78, and the lowest in 2008 with 37.31/100 thousand inhabitants and a mortality rate 6.72. The analysis by quadrennium also finds a higher incidence in 2016–2019 with 42.75/100 thousand inhabitants, with an increase of 7.85% compared to 2008–2011, and the mortality rate increasing from 7.1 to 8.61. The Global Burden of Disease Study 2017 (GBD 2017) argues that CVD mortality rates have decreased significantly in recent decades, however the total number of deaths has increased, probably due to population growth and aging [24].

There was greater involvement in the older adult population with a higher prevalence of deaths in individuals over 60 years of age, especially the group aged 70 to 79 years, responsible for 27.13% of deaths and with a mortality rate of 10.2; however, the group aged 80 years and over (26.79%) had the highest mortality rate 15.69. In analyzing the four-year period, the age group from 20 to 49 years old presented a reduction, while the groups from 50 years old to 80 years old and more presented an increase. These data are similar to the study by Pellense et al. (2021), who also found a greater involvement in the older adult population in the period from 2015 to 2019 [25].

Males showed a higher incidence in the number of deaths (51.20%) compared to females (48.79%), with a mortality rate of 8.08 and 7.75 respectively. In comparing the four-year periods, males showed an increase of 13.63% in the number of deaths between 2008–2011 and 2016–2019, with a mortality rate going from 7.5 to 8.46. On the other hand, females showed an increase of 13.34%, as well as an increase in the mortality rate from 6.87 to 8.39. Studies show that despite the small difference between the percentages of deaths between females and males, it is identified in most results that males have a predominance in the number of diagnoses and deaths [26].

In addition, the following findings stand out regarding deaths and mortality rates: 1) a 6.46% reduction in deaths from HF; 2) an increase in the number of deaths and mortality rate in all regions; 3) a 7.85% increase in deaths between 2008–2011 and 2016–2019, as well as an increase in the mortality rate; 4) a progressive increase in the number of deaths and mortality rate with increasing age, especially in individuals over 60 years of age; 5) an increase in the number of deaths and mortality rate for both genders.

In their analyses, Malta et al. (2019) point to a decline in NCD mortality rates in the first years after implementing the Action Plan, however, there was a worsening in mortality indicators and some risk factors from 2015 onwards, which may compromise the 2% reduction target per year by 2022 [27]. Studies indicate that the worsening of health indicators in the country is related to the country's severe economic and political crisis [28,29]. Other factors such as genetic predisposition, advanced age, inadequate life habits, environmental aspects such as stress and work overload and exposure to pollution are strongly associated with the increase in the number of deaths from CVDs [26].

However, the significant demographic and social heterogeneity existing in the country must be taken into account. The environment in which the individual is inserted reflects on their habits, lifestyle and health conditions [26]. The 2017 GDB study showed that the reduction in prevalence by age occurred unevenly in the Brazilian federative units, being greater in the more developed regions of the country [24]. Furthermore, inequality in the levels of healthcare and promotion due to lack of resources is evident between regions in Brazil [30].

In this study, a reduction in the hospitalization rate of the adult population was observed after implementing the Strategic Action Plan for Coping with CNCDs (2011–2022), however,

there was no improvement in relation to the number of deaths and mortality rate from CVD. This shows that there is still a long way to go to reduce the impact of these diseases in Brazil, and reaffirms the need for and importance of maintaining the prevention of risk factors, the social determinants of health and care reorganization in the face of population aging.

The 2021 annual Statistical Update of the American Heart Association, in conjunction with the National Institutes of Health reports the most up-to-date statistics related to heart disease. The prevalence of CVD (comprising CHD, HF, stroke, and hypertension) in adults ≥20 years of age is 49.2% overall (126.9 million in 2018) and increases with age in both males and females [31]. According to European Cardiovascular Disease Statistics 2017, CVD are the leading cause of mortality in Europe as a whole, responsible for over 3.9 million deaths a year, or 45% of all deaths. In the latest available year, inpatient admission rates in EU countries were 30% higher for males than females on average. Admission rates for acute myocardial infarction were more than two times higher in males than females, while those for heart failure and cerebrovascular diseases were 10% higher in males than females on average [32].

It is important to point out that this study has some limitations regarding the use of secondary data which may suffer registration variations, inadequate data storage and inconsistencies in filling in the causes of death. Population estimates in the country may also be subject to error. However, the databases used in this study are considered official government and are effective in epidemiological monitoring.

## Final considerations

Although there was a reduction in hospital admissions for CVD between 2008 and 2019 in most analyses, the number of deaths observed in the studied period deserves specialized attention. The analysis makes it possible to reflect on the impact of the disease on the health of the Brazilian population, especially in the older age group. Such findings contribute information which allow better control and monitoring of CVDs, and should be taken into account when implementing new strategies for prevention, care and control of risk factors.

The struggle to prevent and improve the population's quality of life is extremely necessary, especially in a developing country like Brazil where resources are scarce. In addition, the importance of maintaining strategy and action plans to combat these diseases is highlighted in order to reduce inequalities in public health. Effective measures, promotion, and prevention for healthcare must be intensified in order to improve the management of the treatment of these diseases and consequently promote a reduction in the rates of hospital admissions and deaths from CVDs.

Some actions foreseen in the plan should be better applied, such as: the intersectoral contribution of the other ministries in the preparation and implementation of the plan of action, improving studies of analysis of morbidity and mortality and surveys, evaluation of health interventions, studies on health inequalities, identification of vulnerable populations, provide a detailed assessment of regional costs with CVDs, promote training of primary health care teams linked to the single health service (SUS), among others. In addition, further studies should be carried out to point out possible strategies for a better impact of the plan in the country.

## Supporting information

**S1 File. Complete statistical analysis.**
(PDF)

## Author Contributions

**Conceptualization:** Rafaella Alves da Silva, Luiza Gabriela de Araújo Fonseca, Lucien Peroni Gualdi, Illia Nadinne Dantas Florentino Lima.

**Data curation:** Rafaella Alves da Silva, Luiza Gabriela de Araújo Fonseca, João Pedro de Santana Silva.

**Formal analysis:** Rafaella Alves da Silva, Luiza Gabriela de Araújo Fonseca, João Pedro de Santana Silva, Illia Nadinne Dantas Florentino Lima.

**Investigation:** Rafaella Alves da Silva, Luiza Gabriela de Araújo Fonseca, Illia Nadinne Dantas Florentino Lima.

**Methodology:** Rafaella Alves da Silva, Luiza Gabriela de Araújo Fonseca, Lucien Peroni Gualdi, Illia Nadinne Dantas Florentino Lima.

**Project administration:** Illia Nadinne Dantas Florentino Lima.

**Supervision:** Illia Nadinne Dantas Florentino Lima.

**Validation:** Illia Nadinne Dantas Florentino Lima.

**Visualization:** Rafaella Alves da Silva, Luiza Gabriela de Araújo Fonseca, João Pedro de Santana Silva, Núbia Maria Freire Vieira Lima, Lucien Peroni Gualdi, Illia Nadinne Dantas Florentino Lima.

**Writing – original draft:** Rafaella Alves da Silva.

**Writing – review & editing:** Rafaella Alves da Silva, Luiza Gabriela de Araújo Fonseca, João Pedro de Santana Silva, Núbia Maria Freire Vieira Lima, Lucien Peroni Gualdi, Illia Nadinne Dantas Florentino Lima.

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
