## [Decision Letter · Decision Letter 0]

8 Apr 2022

PONE-D-22-03798The impact of the strategic action plan to combat chronic non-communicable diseases on hospital admissions and deaths from cardiovascular diseases in BrazilPLOS ONE

Dear Dr. Silva

Thank you for submitting your manuscript to PLOS ONE. After careful consideration, we feel that it has merit but does not fully meet PLOS ONE’s publication criteria as it currently stands. Therefore, we invite you to submit a revised version of the manuscript that addresses the points raised during the review process.

We look forward to receiving your revised manuscript.

Kind regards,

Deepak Dhamnetiya, MD

Academic Editor

PLOS ONE

Journal Requirements:

Reviewers' comments:

Reviewer's Responses to Questions

**Comments to the Author**

1. Is the manuscript technically sound, and do the data support the conclusions?

Reviewer #1: Yes

Reviewer #2: Yes

Reviewer #3: Yes

2. Has the statistical analysis been performed appropriately and rigorously? 

Reviewer #1: Yes

Reviewer #2: Yes

Reviewer #3: Yes

3. Have the authors made all data underlying the findings in their manuscript fully available?

Reviewer #1: Yes

Reviewer #2: No

Reviewer #3: Yes

4. Is the manuscript presented in an intelligible fashion and written in standard English?

Reviewer #1: Yes

Reviewer #2: Yes

Reviewer #3: Yes

5. Review Comments to the Author

Reviewer #1: General comment on the article:

The article showed an important subject which is the impact of strategic planning in the disease prevention and control

In the introduction The authors did not give the details about the strategy such as the goals , the objectives, the indicators that was expected, some activities for the implementation, so the readers know exactly how the strategy was structured , and the result expected from it. This is important if there will be any amendments that should be done.

The article have big data that give real insights on NCDs in the country, the article showed result with stratification by years, age , region , gender and grouping by years and age , the result showed too much details that does not needed to be present in the context but can be shown in tables and graphs, Too much numbers and percentages in the context may make the readers lost, we recommend less numbers and keep the important one and explain if there is trend

The discussion was better than the result in the way it was organized and really showed what the author want to answer , it though needed to be compared with more articles in the region and the world , and the author may give some reasons for why the result was like that (professional opinion based on author observation or other studies in the country)

One limitation of the study is it does not consider any changes in risk factors for NCDs as they change before the changes happened in the hospitalization and the mortality rate , but this may be not available in the data used by the authors

In the last section there should be recommendations to improve the strategy and recommendations for further studies

Reviewer #2: This is an interesting manuscript which highlighted the cardiovascular risk in hospital admission and death. Samples based on secondary data is adequate and statistical analysis is suitable to draw conclusion about the study. Authors tried to explore the data as much as possible but descriptive statistics of hospital admission, death and mortality rate; and the results by two-way ANOVA test should be added by the tables as supporting information to be more clear interpretation.

Reviewer #3: This paper is interesting study on the evaluation of the impact of the strategic action plan to combat chronic non-communicable diseases on hospital admissions and deaths from cardiovascular diseases in Brazil. The study further analysed time trend from 2008 to 2019 and useful in supporting health promotion public policies. I have provided a few suggestions to improve the manuscript below:

In the abstract, Please express HF.

In data extraction, how do you check the data validity and accuracy?

Line 115 - please describe ‘age group (20 to 80 years or older)’.

Line 116 – what does it mean, “The number of hospital admissions and deaths per 100,000 inhabitants was calculated by the ratio between the number of hospital admissions and deaths recorded and the estimate of the Brazilian population, multiplied by 100,000, according to the population projections made by the IBGE.” Please provide clear information about mortality rate.

Line 133- I didn’t find these data in Table 1 “Heart Failure (20.68%), other ischemic heart diseases (13.54%) and stroke (11.98%), Table 1”. Please check it.

Line 142 – What do you mean this sentence “When evaluating the difference between region of residence and year in relation to hospital admissions/100 thousand inhabitants, it is possible to observe that there is a difference for both region (p<0.001)”

Figure 1, 5 and 9 should be describe with Line graph.

In discussion, there is a lot of repetition of the results. It is possible to have a good discussion without putting so many results from the study itself. It is also possible to dialogue with the studies that served as a basis without putting their numerical results. The results of incidence and regional distribution as well as the time trend should compare with other countries.

According to your objective, you should firstly discuss with line 392-

“The main findings in the study with regard to hospital admissions for CVDs after implementing the Strategic Action Plan for Coping with NCDs (2011-2022) were: 1) a 25.77%----------”. Please provide more information why decrease and increase on hospital admissions, deaths and mortality rate.

6. PLOS authors have the option to publish the peer review history of their article (what does this mean?). If published, this will include your full peer review and any attached files.

Reviewer #1: No

Reviewer #2: No

Reviewer #3: No

---

## [Author Response · Author response to Decision Letter 0]

23 May 2022

Reviewer #3: General comment on the article:

1. This paper is interesting study on the evaluation of the impact of the strategic action plan to combat chronic non-communicable diseases on hospital admissions and deaths from cardiovascular diseases in Brazil. The study further analyzed time trend from 2008 to 2019 and useful in supporting health promotion public policies. I have provided a few suggestions to improve the manuscript below:

In the abstract, Please express HF.

Author response: Thank you so much for your comment. However, this acronym does not appear in this section.

2. In data extraction, how do you check the data validity and accuracy?

Author response: The data validity and accuracy were not assessed by statistical tests since the extraction was performed by only one author. Then, the information was checked and verified by a second author.

3. Line 115 - please describe ‘age group (20 to 80 years or older)’.

Author response: Thank you so much for your comment. The suggestion was accepted, and the description was made.

4. Line 116 – what does it mean, “The number of hospital admissions and deaths per 100,000 inhabitants was calculated by the ratio between the number of hospital admissions and deaths recorded and the estimate of the Brazilian population, multiplied by 100,000, according to the population projections made by the IBGE.” Please provide clear information about mortality rate.

Author response: Thank you for your comment. Since the values presented in the DATASUS extraction were raw values of hospital admissions and deaths, we carefully considered the population of each Brazilian region so that the data were proportional. Thus, we improve the sentence to “The number of hospital admissions and deaths was calculated by the ratio between the number of hospital admissions and deaths recorded and the estimate of each region Brazilian population, according to the IBGE.”

5. Line 133- I didn’t find these data in Table 1 “Heart Failure (20.68%), other ischemic heart diseases (13.54%) and stroke (11.98%), Table 1”. Please check it.

Author response: Thank you so much for your comment. The sentence was revised.

6. Line 142 – What do you mean this sentence “When evaluating the difference between region of residence and year in relation to hospital admissions/100 thousand inhabitants, it is possible to observe that there is a difference for both region (p<0.001)”

Author response: Thank you so much for your comment. The sentence was revised.

7. Figure 1, 5 and 9 should be describe with Line graph.

Author response: The suggestion was accepted.

8. In discussion, there is a lot of repetition of the results. It is possible to have a good discussion without putting so many results from the study itself. It is also possible to dialogue with the studies that served as a basis without putting their numerical results. The results of incidence and regional distribution as well as the time trend should compare with other countries.

Author response: The suggestion was very important. We added this information according to Heart Disease and Stroke Statistics, 2021 Update, A Report from the American Heart Association and European Cardiovascular Disease Statistics 2017 edition. 

9.According to your objective, you should firstly discuss with line 392 - “The main findings in the study with regard to hospital admissions for CVDs after implementing the Strategic Action Plan for Coping with NCDs (2011-2022) were: 1) a 25.77%----------”. Please provide more information why decrease and increase on hospital admissions, deaths and mortality rate.

Author response: The suggestion was accepted and so we reformed the information in lines 425-419.

---

## [Editor Report · Decision Letter 1]

25 May 2022

The impact of the strategic action plan to combat chronic non-communicable diseases on hospital admissions and deaths from cardiovascular diseases in Brazil

PONE-D-22-03798R1

Dear Dr Silva,

We’re pleased to inform you that your manuscript has been judged scientifically suitable for publication and will be formally accepted for publication once it meets all outstanding technical requirements.

Kind regards,

Deepak Dhamnetiya, MD

Academic Editor

PLOS ONE

---

## [Editor Report · Acceptance letter]

30 May 2022

PONE-D-22-03798R1 

*The impact of the strategic action plan to combat chronic non-communicable diseases on hospital admissions and deaths from cardiovascular diseases in Brazil*

Dear Dr. Silva:

I'm pleased to inform you that your manuscript has been deemed suitable for publication in PLOS ONE. Congratulations! Your manuscript is now with our production department. 

Kind regards, 

on behalf of

Dr. Deepak Dhamnetiya 

Academic Editor

PLOS ONE